# A Qualitative Evaluation of the Australian Community Pharmacy Agreement

**DOI:** 10.3390/pharmacy11060188

**Published:** 2023-12-18

**Authors:** John K. Jackson, Betty B. Chaar, Carl M. Kirkpatrick, Shane L. Scahill, Michael Mintrom

**Affiliations:** 1Faculty of Pharmacy and Pharmaceutical Sciences, Monash University, Parkville, VIC 3052, Australia; 2School of Pharmacy, University of Sydney, Sydney, NSW 2006, Australia; betty.chaar@sydney.edu.au; 3School of Pharmacy, University of Auckland, Auckland 1023, New Zealand; 4School of Social Sciences, Faculty of Arts, Monash University, Melbourne, VIC 3800, Australia; michael.mintrom@monash.edu

**Keywords:** community pharmacy, policy, funding, scope of practice, medication management

## Abstract

The Australian Federal Government’s Community Pharmacy Agreement (Agreement), initiated in 1990 and renegotiated every five years with a pharmacy owners’ organisation, is the dominant policy directing community pharmacy. We studied the experience with the Agreements of 38 purposively selected individual pharmacists and others of diverse backgrounds, using in-depth, semi-structured interviews. Although perceived to lack transparency in negotiation and operation, as well as paucity of outcome measures, the Agreements have generally supported the viability of community pharmacies and on balance, contributed positively to the public’s access to medicines. There were, however, contradictory opinions regarding the impact of the policy’s regulation of pharmacy locations, including the suggestion that they provide existing owners with an undue commercial advantage. A reported shortcoming of the Agreements was their impact on pharmacists’ abilities to expand their scopes of practice and assist patients to make better use of medicines, in part due to the funding being almost totally focused on supply-related functions. The support for programs such as medication management services was perceived to be limited, and opportunities for diversification in pharmacy practice appeared constrained. Future pharmacy policy developed by the government could be more inclusive of a diverse range of stakeholders, seek to better utilise pharmacists’ expertise, and have a greater focus on health outcomes.

## 1. Introduction

The legislative framework for community pharmacy in Australia is divided between the (i) state and territory governments and (ii) federal (national) government. State and territory governments have a primary responsibility for the profession through their registration of pharmacists and the regulation of their practises. They also regulate pharmacy premises, limit their ownership to pharmacists, and determine the number in which each pharmacist can have a pecuniary interest [1]. However, the federal government has a responsibility for funding medicines and pharmacists dispensing of them, which provides this level of government with significant influence. The federal government’s funding model, the Community Pharmacy Agreement (CPA) has for many years been the dominant policy affecting community pharmacy (CP) [2,3].

The CPA regulates and remunerates pharmacies for dispensing medicines and other services under the Pharmaceutical Benefits Scheme (PBS), the government-operated pharmaceutical insurance program, which is an integral part of the country’s universal health coverage. In 2021–2022, the PBS cost the federal government and public combined AUD 16.3 billion, of which 28.5% was expended on the supply chain, principally pharmacies [4]. 

Initiated in 1990 as a result of negotiation with the Pharmacy Guild of Australia (PGA), an industrial organisation representing a portion of pharmacy owners, and renegotiated every five years almost exclusively with the PGA, the seventh iteration of the CPA was scheduled to operate from 2020 to 2025 [5,6]. The stated purpose of the CPA found in the objectives of all seven Agreements include supporting a viable community pharmacy network, ensuring the public’s access to medicines, and delivering quality services [5,6]. The references to ‘quality’ include quality patient care outcomes (1CPA), quality pharmacy services (4CPA), and the quality use of medicines (QUM) (6CPA). Operational factors of transparency, accountability, efficiency, and effectiveness are also listed as objectives in various Agreements; however, these terms are not defined in the Agreements. 

Two programs have been the central pillars of all seven Agreements. These are the funding of dispensing and pharmacy location rules, both of which align with the objectives of maintaining a viable pharmacy network and ensuring public access to medicines. The initial CPA also included structural reform of the sector, in that it funded pharmacies to close or amalgamate so as to improve the distribution across the country [7,8]. Linked to this, rules were introduced to regulate the location of new PBS-approved pharmacies and the relocation of existing approved pharmacies, including to prevent backfilling into locations where pharmacies had been funded to close. In effect, the rules which have been maintained in a modified form in all subsequent Agreements prevent pharmacies from relocating to more densely populated areas or more viable locations, at the expense of other locations. Approximately 10% of pharmacies closed during the 1CPA [2] and as the location rules have constrained the opening of new pharmacies, the 5600 that existed in 1990 grew only slightly to 5901 by 2021–2022 [4,9].

### Other CPA Programs Have Included the Following:

Community Service Obligation (CSO) payments to nationally operating full-line pharmaceutical wholesalers to hold and deliver essential medicines to pharmacies within 24 h (4CPA to 7CPA) [10];Research and Development (R&D) funding of approximately AUD 95 million over five CPAs principally to support the development of pharmacy practice through new Professional Programs (2CPA to 6CPA) [11];Funding of Professional Programs including medication management services and medication adherence services (2CPA to 7CPA). In 2021–2022, the expenditure on professional programs of AUD 242.8 million accounted for less than 1.5% of total PBS expenditure (Table 1) [4].

**Table 1 pharmacy-11-00188-t001:** Professional Programs expenditure 2021–2022 [4].

Category	Program	Expenditure (AUD in Millions)	Percentage of PBS exp.
Medication management services (MMS)	Residential Medication Management Reviews	13.25	
	Home Medication Reviews	22.00	
	MedsChecks (MC) and Diabetes MC	40.0	
	Quality Use of Medicines	8.00	
Total MMS		83.25	0.50%
Medication adherence services (MAS)	Dose Administration Aids (DAA) and Indigenous DAA	113.00	
	Staged Supply	8.70	
Total MAS		121.70	0.73%
Aboriginal and Torres Strait Islander programs	Various	3.94	0.02%
Rural Pharmacy Maintenance Allowance		21.00	0.13%
Rural Pharmacy Workforce Program	Various	4.87	0.03%
Program Administration		8.10	0.05%
Professional Program total		242.86	1.45%
Total PBS expenditure		16,700.00	100%

While it has provided stability for the government and support for pharmacy proprietors, aspects of the CPA have been criticised by the Australian Healthcare and Hospitals Association [12], the Competition Policy Review [13], the Consumers Health Forum [14,15], the Australian National Audit Office [16], the Grattan Institute [17], the Productivity Commission [18], and the Pharmaceutical Society of Australia (PSA) [19], amongst others. The issues raised have included the following: the inequity of the pharmacy location rules [12], lack of transparency regarding costs and outcomes [12], the CPA’s anticompetitive nature [13,18], lack of health outcome measures [15], lack of consumer engagement [14], poor administration [16], and constraint on the development of pharmacists’ practices due to the limited support for professional programs [17,19]. 

A set of measures for the 7CPA addressing legal compliance, funding, and processes but not outcomes has been set by the government [20], and yet the government’s 2022 review of the 7CPA stated that a key issue for the Agreements has been “the scarcity and quality of available data for robust and meaningful analysis of health outcomes” [21].

Prior analysis of the CPA as a government funding Agreement found that “a powerful private business interest, the PGA, played a central role in initiating the CPAs and has been key to their evolution and maintenance through regulatory capture” [22]. A second evaluation using policy theory characterised the CPA as predominantly being an industry policy [23]. Although the CPA has been cited frequently in research into CPs in Australia [24,25], no comprehensive analysis has been undertaken of the experience of stakeholders with the Agreements. 

Hanberger argues that when a policy is instigated, it is unknown how stakeholders with different and potentially conflicting interests will behave, how the policy will evolve, and what consequences will arise [26]. In 2023, the government unilaterally imposed a change in the dispensing frequency for some PBS medications, which the PGA alleged would affect the profitability of pharmacies. In response to the change in the government policy, the PGA lobbied successfully for the immediate negotiation of a new Agreement, with a proposal for an 8CPA to commence in 2024 [27]. 

In spite of the criticisms levelled at the CPA including the claim of ‘regulatory capture’ by the PGA and the recent dispute between the government and the PGA, it remains the dominant policy in CP. We believe this anomalous position of the CPA warrants a detailed analysis. The aim of this research was to study the experience and perspectives of the CPA, held by a range of stakeholders who influence, apply, or are affected by the policy framework of CPs in Australia, so as to inform future policy development. The objectives were as follows:Evaluate whether the principal objectives of the CPA have been met;Examine the performance of key functional aspects of the CPA;Assess the outcomes of the CPA, particularly in regard to the development of pharmacists’ practices;Consider options for future government–profession contractual arrangements.

## 2. Methods

This study utilised a qualitative analysis of semi-structured interviews with stakeholders, supported by the analysis of primary CPA documents and PBS and CPA expenditure data [4,5,6]. 

Ethics

The study was approved by the Monash University Human Research and Ethics Committee (ID: 31875).

Recruitment

Stratified purposive sampling [28] was used to select individuals in approximate balance from four socioecological strata of CP [29]: societal, community, organisational, and personal strata. Entities and individuals constituting each stratum are presented in Table 2. Presidents of major pharmacy and nonpharmacy organisations and other key opinion leaders in the public domain were invited to participate. Snowball sampling was used when necessary to ensure the inclusion of individuals from all substrata and balance between the strata and across the nation [28]. Contact was initially made via email and included an explanatory statement and consent form. If no response was received, a follow-up was performed via email or telephone. 

Data collection

Semi-structured interviews, designed to be approximately one hour in duration, were conducted at a date, time, and place convenient to the interviewee, online (Zoom) or face-to-face, between December 2022 and April 2023. All interviews were conducted by the lead author (JJ), a PhD candidate with experience in qualitative research methods including semi-structured interviews. A phenomenological approach was applied in the development of the interview guide (Appendix A Interview Guide) which was informed by the literature [30] and the objectives and programs of the CPA. Pilot interviews were conducted with experienced pharmacists and the guide was modified as necessary. Confidentiality was maintained throughout, with interviewees identified by a code number in all field notes, interview transcripts, and data analyses. 

Data analysis

Audio recordings of the interviews were professionally transcribed verbatim, and the transcripts were checked for accuracy against the audio recordings and notes taken during interviews. Following data cleaning, familiarisation was undertaken by rigorous reading of the transcripts, with key concepts identified using deductive thematic analysis [31], and codes were selected, described, and assigned by the lead author using the framework of Pope et al. [31]. The transcripts were closely reread to identify secondary codes, which were reviewed with the other researchers and discussed until consensus was reached that all codes were coherent, distinct, and linked to the research objectives. A coding tree (Appendix B Coding Guide) was created in NVivo (QSR International, university licence) [32]. On further readings of the transcripts, relevant text segments were mapped to the codes, and illustrative quotes were identified [31].

To mitigate bias, an independent researcher (BC) experienced in qualitative research but not involved in setting the research objectives or conducting the interviews was engaged at this stage, specifically to objectively collaborate in curating the data and analysing the results. The objectives, codes and code descriptions, relevant transcripts, selected text segments, and mapping by the lead author were provided to BC for validation. Following discussion and resolution of variations, the data were reviewed by the other authors.

The COREQ checklist was adopted as a methodological guide [33].

## 3. Results

Forty-nine individuals were approached to participate in the study of which seven failed to respond, four refused, and two provided referrals to other suitably relevant individuals. The subsequent 38 interviews were of an average duration of 46 min (range of 26–80 min), with individuals from seven of the eight Australian states and territories, and included 24 pharmacists (including members of the PGA) and 14 women. Interviewees included nine representatives of pharmacy organisations, nine politicians, regulators, and policy experts, seven pharmacy proprietors, seven community pharmacists or pharmacists engaged in other sectors, seven consumers or consumer (patient) representatives, and six individuals from nonpharmacy stakeholder organisations. The number exceeds 38 as some participants were members of more than one stratum. No repeat interviews were required. The socioecological strata of the participants are presented in Table 2. Interviewee anonymity codes, areas of responsibility, and expertise are in Appendix C Interviewee Codes.

Themes emerging from the data that are reported below include the four objectives of the CPA: access to medicines, viable CP network, quality, and transparency. This is followed by four functional aspects of the CPA including the negotiation process, funding, location rules, and Professional Programs. Finally, three outcomes of the CPA, the actual beneficiaries, impact on practice development, and future, are addressed.

### 3.1. Objectives of the CPA

The consensus of the interviewees was that the Agreements lacked a vision for CP’s role in health care and were focused on funding rather than health objectives. 


*My impression is, in many cases the objective was to get an agreement, rather than to have a real vision for what pharmacists and medication can do to improve the health of the Australian population.*
(18/Policy)

#### 3.1.1. Access to Medicines

Opinions were varied as to whether the CPA’s objective to support the public’s access to medicines had been met. It was suggested to have “*done a relatively good job*” (19/Prop/Accredited), “*even in time of crisis like flood and bush fire*” (12/Policy), while another stated that they “*don’t think that it* (support for access) *is proven”* (24/Policy). While the public’s access to medicines is multifactorial, it is essentially underpinned by the PBS listing and subsidisation of essential medicines [34]. Some interviewees conflated ‘access to medicines’ with ‘access to pharmacies’, stating that the CPA’s support for a viable pharmacy network ensured access to medicines and contrasted ready access to pharmacies with difficulties accessing medical practitioners, particularly in rural settings. 


*I think the CPAs have managed to maintain equity of access much better than has been the case with things like doctors’ surgeries, which are an example of something that was deregulated around about the same time as the CPA came in.*
(9/Prop)

Other respondents were more specific, suggesting that the CPA’s impact on access was limited to the Community Service Obligation’s (CSO) funding of wholesalers to deliver medicines to pharmacies within 24 h, thereby enabling them to be dispensed in a timely manner. The CSO was reported to particularly benefit patients of small, rural, and remote pharmacies. 


*We have wholesalers who deliver *(medicines)* in 24 h no matter when. They’re the ones that actually ensure everyone in the country has access to a PBS medicine, not pharmacists.*
(26/Bureaucrat)

On the other hand, the CPA’s location rules were perceived by some to limit the distribution of pharmacies, thereby impeding the public’s choice of pharmacy and their access to medicines. 


*There is a laundry list longer than my arm of rural, regional and even suburban locales that have multiple citizens’ petitions seeking *(XYZ pharmacy group (Adjusted to maintain anonymity))* to open in their community and we have no response other than to say sorry we can’t–law does not permit–that’s not providing equitable, timely, universal access to PBS!*
(37/PhAssn/Prop)

Finally, others suggested that the CPA was irrelevant to access because medicines were accessible from pharmacies prior to the CPA and would still be available without an Agreement. 


*Even if the CPA weren’t there, the pharmacies would still be there and there’d probably be more of them.*
(8/Phcist)


*Well I don’t see why …. access to PBS medicines would be any different without a CPA.*
(5/PhAssn)

#### 3.1.2. Viable Network 

Between 1989/90, and 2021/22, the net number of pharmacies increased by 5.4%, while the Australian population grew by 54% [35], and the resultant population to pharmacy ratio increased from 3000:1 to 4426:1 [18]. The numbers of PBS-subsidised prescriptions increased by 105%, and the combined government and patient payments for subsidised prescriptions increased by 1076% [4]. While margins narrowed, it was argued that the dispensing revenue supported by these demographic changes underpinned the viability of the overall CP network.

*The answer* (to the question of a viable CP network) *in rural and remote Australia is: yes. By and large yes, I think it is *(viable)*. In metropolitan I think there is over-funding, so it’s created an opportunity for excess profits.*(18/Policy)

It was stated that the viability of individual pharmacies was influenced by commercial factors. Some pharmacies are members of marketing (banner) groups and while the four largest groups account for approximately 30% of pharmacies, they account for 73% of the market [18]. Indeed, the success of volume- and price-based pharmacy groups was contrasted with and seen as negatively impacting smaller, service-focused pharmacies by a number of respondents. 


*The CPA was successful in delivering monopolies and the location rules have stopped innovation.*
(14/Consumer)


*It suppresses the striving for greater quality, for greater improvement in the provision of services, a greater focus on the consumer.*
(17/Policy)


*The current program, this is all about …. how many medicines you hand over as opposed to if you’re genuinely moving into primary care and looking for a different scope of practice.*
(26/Bureaucrat/Consumer)

#### 3.1.3. Quality 

The Quality Care Pharmacy Program (QCPP) which was initiated through the CPA, was seen as solely focused on the commercial aspects of pharmacies, not relevant to the professional practice, and having marginal benefit. Home Medication Reviews (HMR) and Residential Medication Management Reviews (RMMR) were the CPA programs most commonly cited as enhancing the quality use of medicines, while many participants stated that there is no evidence that the Agreements have done anything in relation to quality. 


*If by quality, you mean the quality use of medicines, then I would say it may have at the margins, but at great cost, and with very little accountability as to whether that has been successful.*
(38/PhAssn/Prop)

The Agreements do not include quality measures, audit functions, or funding for the assessment of quality, and these deficiencies were cited as impediments to expanding pharmacists’ roles. 


*Metrics that we would value as being measures of quality use of medicines and the appropriate indicators are lacking. … services such as HMR and RMMR’s, dose administration aids and MedsChecks, we count them but we don’t necessarily measure their impact.*
(7/Assn)

CPA-related quality cannot be divorced from the broader questions of quality in community pharmacy. Two non-pharmacist respondents identified the promotion of nonevidence-based medicines in CP as reflecting poorly on the quality of pharmacists’ practices.


*Pharmacists, like nurses and doctors, are trusted clinicians so if they are propagating mis-information about what actually works and doesn’t, then we’ve got a real problem in terms of helping the community to remain health illiterate.*
(22/Assn/GP)

#### 3.1.4. Transparency 

There is no guidance in the CPA documents as to the intent of the ‘transparency’ objective or how the objective is to be assessed, although the performance measures established for the 7CPA provide limited transparency [20]. When asked about the transparency objective, most interviewees addressed the transparency of CPA negotiations. Representatives of non-PGA stakeholders reported inadequate transparency and little value in pre-negotiation consultation, although one recognised “*working in the policy environment, there’s always a trade-off between getting things done and being transparent*” (34/Phcist/Consumer). An economist argued that negotiating the CPA was ‘rent seeking’ by the PGA and “*those who are seeking rent from the Commonwealth—the last thing they want is transparency and bluntly the Commonwealth …. it doesn’t want transparency either*” (24/Policy). 

Interestingly, the majority of association representatives, nonproprietor (employed) pharmacists, and consumer representatives claimed that a lack of transparency continued during the five-year period of each CPA, particularly in relation to overspending, underspending, limitations on services, and the allocation of money to specific programs.


*I challenge governments and bureaucrats to look at which other government program is worth $18 billion over 5 years, and has … this *(low)* level of transparency and accountability and reporting. I think you’d struggle to find any programs—not just in health, I’m talking across all, across all government sectors.*
(6/PhAssn/Policy)

### 3.2. Functional Aspects of the CPA

The four functional aspects of the CPA discussed were as follows: the negotiation process, funding, location rules, and professional programs.

#### 3.2.1. The CPA Negotiation Process

The contrasting objectives of the government and the PGA need to be recognised in assessing CPA negotiations. The representatives of the bureaucracy and consumers stated that the government has a responsibility to ensure that the money is spent on the CPA effectively and transparently, and that outcomes are delivered. On the other hand, participants unanimously recognised the PGA as an industrial organisation, with a responsibility to negotiate arrangements advantageous to the financial interests of its members. 


*As a pharmacist having the CPA negotiated by a small group of owners with a vested interest is quite insulting, and quite inappropriate. They have a vested interest in achieving commercial prosperity.*
(10/PhAssn/Accredited)

The PGA’s strength as a political lobby was cited by many as underpinning its significant role in CPA negotiations, and its political donations were raised by two respondents as a contributing factor. Many interviewees perceived that professionalism and healthcare were secondary priorities to the negotiation of financially acceptable outcomes. 


*No health minister, or treasurer for that matter, is willing to use the government’s PBS monopsony to negotiating advantage. I think they throw it away; they don’t really want the confrontation, they want to let sleeping dogs lie, they don’t want to antagonise the Guild and they don’t want to have Guild pharmacists campaigning on high streets against them at election time.*
(17/Policy)

A two-party negotiation was seen by most as an unsatisfactory way for the government to engage with the multiple stakeholders of the profession, as “*no one is educated enough on other’s interests to advocate on their behalf*” (37/PhAssn/Prop). The involvement in the 7CPA negotiations of the professional body, the Pharmaceutical Society of Australia (PSA), was universally seen as overdue, while some thought that it was inadequate.


*PSA probably should get a little bit more of a say than just a single clause.*
(15/PhAssn)

The consolidated opinions of the respondents as to who should be at the negotiation table included greater involvement from the PSA as the body setting professional standards, employed pharmacists whose work delivers the services, and consumers as the theoretical beneficiaries of the Agreements. 


*The Australian government invests in health outcomes for consumers, not profits for the industry. The CPA needs to be negotiated in that context aligned with other government agreements.*
(7/Assn)


*As a taxpayer, we are paying for these products and services. The consumer must have input, rather than having it occur behind closed doors.*
(10/PhAssn/Accredited)

Less frequently proposed were other stakeholders such as state and territory governments as regulators of the practices and premises, and hospital pharmacists who dispense 25% of the PBS by value.

The long-term sustainability and stability of the quinquennially-negotiated CPAs drew contrasting opinions from the interviewees. A point made by proprietors and some others is that the time frame of the CPA is inadequate to support investment, as it does not provide certainty for funded programs, and “*there needs to be a mechanism… to lock in a lot of this funding to ensure sustainability*” (11/PhAssn/Prop). This point is exemplified by the number of professional services introduced and subsequently ceased (Table 3).

The negotiations, described by a non-pharmacist as “*battles royal”* (24/Policy) were cited as a wasteful drain on both parties. The consensus was that any agreement involving billions of dollars of public money should be established in a different way and with a clearer scope. As the CPA has existed for over 30 years, most pharmacist interviewees have no experience in alternative funding frameworks, whereas policy experts and commentators external to the profession proposed other mechanisms to set an appropriate fee, such as through independent arbitration and contracting. 

#### 3.2.2. Funding

Of those who claimed to understand the funding model, many stated that it rewards the speed and volume of dispensing, rather than the patient care or health outcomes. 

(The CPA) *favours the cheap and cheerful, and fast and furious and those that are trying to uphold the high standard actually are struggling to flourish.*(22/Assn/GP)

A financial advisor to the sector stated that dispensing remuneration is adequate for supply-focused business models, whereas many pharmacists identified the single, fixed dispensing fee to be inadequate for complex patients and higher-risk medicines, both of which require extra counselling time. Others stated that it does not reward the provision of comprehensive clinical services or provide incentives for pharmacists to apply their full scope of practice. 


*Some patients require extra time, and that’s not being adequately remunerated within the set dispensing fee. So, I think a lot of pharmacies are using MedsChecks as an extended counselling funding tool.*
(30/Prop/Accredited)

Proprietors perceived their high dependence on the federal government for revenue to be a business risk, and there was some support for the diversification of income by charging patients additional fees. Consumers and some other interviewees thought the latter to be inequitable and contrary to the nation’s universal health coverage, while others suggested that many pharmacists would be reluctant to do so. 


*You don’t want it to become a system where only the wealthy are going to be able to afford to walk into a pharmacy.*
(20/Consumer)


*I think the majority of pharmacists wouldn’t like the latter because you know patient costs are very delicate.*
(36/PhAssn/Policy)

A number of respondents raised the new pharmacist services that state and territory governments are approving under a full scope of practice, such as pharmacists’ initiations of antibiotics for simple urinary tract infections [36,37]. In the absence of funding from the state and territory governments, pressure will mount on the CPA to support these services.


*Increasing scope of practice is excellent and needed, however I don’t think the remuneration model is keeping pace.*
(21/Prop)

A rural pharmacy proprietor cited the Rural Pharmacy Maintenance Allowance [38] zone-based funding as beneficial, although an unexpected change to zoning that occurred coincidental to the purchase had adversely affected their business value. Respondents experienced in health planning and financing suggested that using incentives to establish pharmacies and promote enhanced services in rural and remote areas should be linked to the health needs of specific catchment areas, rather than be zone-based funding. 

#### 3.2.3. Location Rules

Avoiding maldistribution of community pharmacies throughout Australia was seen as an appropriate objective, but distance-based location rules were seen by many as problematic, leading to anomalous outcomes and creating economic distortion. Some interviewees stated that the rules have reduced competition between pharmacies which has decreased the business risk, thereby increasing the valuations. This, along with limited opportunities to establish new pharmacies, has created barriers for young pharmacists seeking to enter and older proprietors looking to exit pharmacy ownership. 


*There’s a lot more demand for pharmacies than what there is supply and because of that people are paying a premium, but they’re paying a super-premium.*
(16/Commentator)


*This particular device has served baby boomer *(older generation)* pharmacists like you wouldn’t believe, except that they are now locked into this golden cage.*
(24/Policy)

While contributing to the consolidation of ownership and capital, it was reported that the changes to distance specifications and the exceptions created for shopping centres undermine the credibility of the location rules. Rather than distance- or shopping centre-based rules, population density and geospatial mapping of access to health services were suggested as ways to achieve better distribution in both metropolitan and rural settings, and ministerial discretion should be applied more frequently in cases where there is a clear public need. Others argued that the location of pharmacies should be left to ‘the market’.


*Because pharmacies are protected by the location rules it makes them lazy because there’s an assumption that everything’s okay and I don’t have to do anything special, I just sit here waiting for the customers to come in.*
(16/Commentator)

Some respondents supported location rules on the basis that they protect pharmacies in rural areas. However, distance-based rules that may prevent competing pharmacies from opening in one-pharmacy towns were reported as creating trapped customer bases, left with no ready choice in relation to the service range or quality. On a wider scale, a pharmacist who has worked widely as a locum in the country stated that the rules have, in some settings, enabled the shared ownership of pharmacies in adjoining country towns to create “*sheltered workshop pharmacy territories*” (35/Phcist*)*. 


*It sets up the cartel model of pharmacy in my opinion and that is not fit for purpose.*
(3/Phcist)

The respondent from the bank sector (32/Commentator) confirmed that the protection from competition and reduced risk flowing from location rules supports increased business valuations. Consequently, bankers lend more generously to pharmacy proprietors than to other small businesses. This advantage would be reduced in the absence of market protection, and many respondents noted that removing the benefit from existing proprietors would be politically challenging.


*Rules based on distance …… really ought to be reviewed. I know people who are established would hate that but it really is unfair to people who want to enter the market, legitimately enter the market.*
(8/Phcist)

Contrasting opinions were expressed in relation to a prior federal government’s decision to legislate the existence of location rules rather than maintain them as a matter that could be used in negotiation with the PGA. On one hand, this change was seen as inconsequential, but on the other, it removed a significant bargaining advantage previously held by the government.


*Rolling over the location rules—which were built in with a sunset clause—was something that we were able to offer which was of enormous value to them *(pharmacists)*, but didn’t cost the Commonwealth anything.*
(2/Politician)

(The government) *gave away the only thing that ever used to be the control point, … the sunset clause in the location rules which is why the Guild would come to the table and offer things up that might be better, *(and)* that the government and patients might actually need.*(26/Bureaucrat/consumer)

#### 3.2.4. Professional Programs

The statement that programs are “*tacked onto the end of the Agreement as a bit of window dressing*” (1/Politician/GP) reflects the opinion of many respondents. 

The current CPA professional programs are listed in Table 1. Many respondents argued that medication adherence services (MAS), including the packing of medicines into dose administration aids (DAAs) and staged supply, should not be classified as professional programs; rather, they should be funded as aspects of dispensing. It was also proposed that rural and Aboriginal and Torres Strait Islander programs should be funded externally to the CPA, with this statement endorsed by both Indigenous interviewees. Others, including consumer representatives, argued that all professional programs should be funded as components of other government programs rather than in the CPA.


*There are lots of tails hanging off the CPA outside of the PBS funding envelope which should not be there.*
(37/PhAssn/Prop)

A subset of the CPA professional programs is professional service, defined as the application of specialist knowledge by a pharmacist with a patient, population, or other health professional, to optimise care and improve health outcomes and the value of healthcare, predominantly through the quality use of medicines [39,40].

Respondents mentioned the lack of transparency on how the research program to develop professional services had been managed, and stated that the continuity and funding of services had been inconsistent. In spite of research, development, and training that supported at least 18 services being introduced (Table 3), many lasted just one Agreement and just four remained in the 7CPA; RMMR, HMR, MedsChecks, and Diabetes MedsChecks. 

Tong et al. [24], Buss et al. [25], and others suggest that professional services have become common; however, interviewees argued that, of those services that do exist, very narrow service scopes had constrained innovation, and the delivery had been impeded by limited funding, inadequate support for travel, and caps which continue to be applied to some services. 


*There was an ability for someone to earn a living outside of the four walls of pharmacy, and obviously with caps …. that limits your ability.*
(5/PhAssn)


*If you look at the most recent community pharmacy agreement, much of the determination is about the price that would be put on a medicine. We’d rather flip that to the value of the service that the pharmacist provides.*
(7/Assn)

Pharmacists engaged in or responsible for delivering services said that they are delivered as episodic events with limited coordination and no outcome measures. They questioned the government’s willingness or capacity to monitor the services, and deemed stronger controls and accountability are necessary to address commodification and to limit overservicing.


*Quality, effectiveness and efficiency, you know, the way that the medication review services are funded under the CPA don’t meet those three elements.*
(6/PhAssn/Policy)

Proprietors emphasised that not all pharmacies are able or willing to deliver a full suite of professional services under the current funding arrangements, due to the required investment in training and consulting rooms and the need for a supporting business model. Others stated that this argument is well illustrated by the limited professional services provided by ‘discount pharmacies.’

(Professional services) *are a cost centre. You’re not paid enough for it to be a profit centre.*(38/PhAssn/Prop)

On the other hand, employed pharmacists said that it was challenging to sustain enthusiasm for providing these services when funding constraints have led to pressure from proprietors to perform services quickly, and the income generated was rarely shared with them. 


*These expanded roles are not going to go anywhere if we don’t fund them properly, because pharmacists will get jack of doing it.*
(27/Phcist/Accredited)

The professional services environment is further complicated by the growth of non-CPA services such as vaccination programs and the initiation of antibiotics for simple urinary tract infections [36,41], which has led to conflict with the medical profession. While medical organisations have opposed some professional services [42], the relationship with medical practices at the local level is more nuanced. One rural pharmacist reported delivering professional services to fill gaps in care arising from the difficulty patients face in consulting doctors in their town, while another in a similar setting elected to limit the delivery of professional services so as to not undermine the viability of their town’s struggling medical practice. 

### 3.3. Outcomes of the CPA

#### 3.3.1. Actual Beneficiaries 

Based on interviewees’ responses, three groups were major beneficiaries of the CPA. Community pharmacy owners and the owner representative organisation of the PGA were cited by 83% of interviewees as gaining the greatest benefit and were frequently identified as the only beneficiaries. The federal government was cited by 33%, and consumers (including patients) were cited by 25% of interviewees as benefitting. Other groups including wholesalers, manufacturers, and other pharmacists were identified less frequently (the total exceeds 100% as some interviewees identified more than one beneficiary). 

While all proprietors were perceived to benefit, specific reference was made to proprietors who own multiple pharmacies, as location rules provide them protection from competitors. 


*Proprietors, and in my language, pharmacy service providers over the age of 55, who have been involved for ownership longer than 20 years and most particularly who have had multiple sites *(are the greatest beneficiaries)*.*
(4/Regulator)

#### 3.3.2. Impact on Practice Development

Many agreed that the dispensing-focused funding model which supported pharmacy viability failed to drive higher levels of clinical responsibility or performance. Due to it preferencing speed of dispensing rather than outcomes and its limited funding for professional services, it was argued that the CPA has promoted a commoditised, transaction-based model of pharmacy. As a result, discount pharmacies have flourished and become internal disruptors, undermining the emergence of an expanded scope and greater professional practice. 


*I don’t think it has progressed pharmacy as far as it should, and my overall impression is that it’s there to make money for pharmacy owners.*
(27/Phcist/Accredited)

Other factors raised as affecting the practice development were the CPA’s constraints on innovation and the delivery of healthcare in a retail environment, with a representative of a medical association suggesting that the retail environment of CPs affected the public’s perception of pharmacists as clinicians. 


*Look at other health professionals. Most of them are independent operators, they get paid for their services directly. Get out of this retail environment … that’s always been the big conflict from my point of view, being in a retail environment.*
(28/Phcist)


*There is confusion about where community pharmacists lie in terms of consumers or patients going into a pharmacy, whether it is a retailing environment or a professional environment.*
(30/Prop/Accredited)

#### 3.3.3. The Future

While a number of respondents stated that the CPA is unlikely to change because of the resistance of vested interests, many argued that the government’s funding arrangements for pharmacy must be changed as a matter of priority. 


*I think it is a very mischievous set of policies that really are no longer fit for purpose and if they maintain them beyond 2025, I think that indicates that our opportunity for a new health system is going to be lost.*
(22/Assn/GP)

Various individuals argued for pharmacy funding to be adapted to contemporary practice, have a greater focus on professional services, include pharmacists in settings other than community pharmacies, be patient centred, focus on medication safety, quality, consumer literacy*,* and medication reconciliation on discharge from hospital, and be integrated and evolve with the developments in healthcare and wider social and technological developments. Respondents across all strata echoed the call for “*more openness, transparency and accountability* and *value for money for our health dollar*” (1/Politician/GP).

The need to change the funding model to enable innovation was raised by many respondents, adding that if there is to be a future CPA, it should be focused solely on remuneration for dispensing, exclude all professional programs, and be established with input from all relevant stakeholders.

A number of interviewees stated that pharmacy proprietors must be compensated for the commercial costs of operating a business such as holding inventory, while many more suggested that remuneration for dispensing should be varied based upon clinical risk. However, a widely reported concern was the likelihood of a centralised fulfilment model disrupting medicine distribution, particularly when integrated with online prescribing services, thereby threatening the existence of brick and mortar pharmacies.


*The uber-fication of our sector is something that will be very interesting.*
(6/PhAssn/Policy)


*The whole reason you would have a pharmacist is being short circuited.*
(29/Assn)

Many respondents argued that the funding for professional programs should be incorporated into other government funding programs. They cited the Medicare Benefits Schedule (MBS) [43], which funds the majority of primary health care practitioners, as a means of funding current professional services plus other standards-based non-dispensing professional activities. These activities ranged from “*over the counter consultation, more complex like a HMR, up to and including embedded pharmacists in aged care*” (10/PhAssn/Accredited). They stated that this would avoid the uncertainty of five yearly negotiations, while paying individual pharmacists directly via the provider numbers would create flexibility in the delivery. 


*At the moment it’s a one size fits all approach. It doesn’t leave a pharmacist as a clinician a lot of ability to exercise professional discretion on how and what level of services will meet the local needs.*
(11/PhAssn/Prop)

Other modifications proposed include rural support being part of the health workforce funding, and pharmacy services supporting Indigenous people to be incorporated into other Indigenous health care delivery. 

## 4. Discussion

We aimed to study the objectives, performance, outcomes, and future of the CPA through the experience and opinions of key stakeholders, using a deductive thematic analysis of the data gathered from 38 semi-structured interviews. 

The data indicated a lack of transparency in the negotiation and conduct of the CPA, and no criteria for assessing the outcomes. However, based on the interviews, the CPA has achieved one of its objectives: supporting a network of viable community pharmacies, with existing pharmacy proprietors being the primary beneficiaries of the Agreements. Due to a number of factors, including restrictions on the establishment and relocation of pharmacies, it was less clear as to whether the network has contributed positively to the public’s ‘equitable, timely, safe and reliable access to medicines’, which is a pillar of the federal government’s National Medicines Policy [44]. 

Dissatisfaction was expressed with the quantum and structure of pharmacy remuneration negotiated under the CPA, with over 98% of PBS expenditure remunerating the dispensing of medication and favouring volume-based pharmacy business models over professional service-based models. However, even if the CPA supports public access to pharmacies and dispensing is viable, there is no evidence that it supports quality, specifically the ‘quality use of medicines and medicines safety’, which is a second pillar of the federal government’s National Medicines Policy [44], and in 2019 was declared the 10th National Health Priority [45]. 

### 4.1. Professional Services

There is strong evidence that pharmacists’ professional services, such as medication reviews, can support the appropriate use of medicines [46,47]. Pharmacists’ abilities to expand their practice scopes and deliver such services, was perceived to be constrained by the limited introduction and evolution of professional services in the CPA, their reported lack of sustainability, underfunding, and the caps applied to specific programs.

Notably, the federal government’s 2023 scope of practice review entitled *Unleashing the Potential of our Health Workforce* has a goal of “*looking at the evidence on health professionals working to, or being prevented from, exercising their full scope of practice in primary care*” [48]. Pharmacists are one of the six groups of primary care practitioners specifically mentioned in the review, implying that it is in patients’ best interests if policy better supports pharmacists to work to their full capacity and scope of practice. 

System-wide benefits beyond the quality use of medicines can accrue from pharmacists working to their full scope of practice. For example, addressing the overwhelming demand on hospitals is impeded by the delays in patients accessing general practitioners, which could be addressed by a greater utilisation of pharmacists [49]. Policy changes enabling pharmacists to address medication-related issues, for example, in residential-aged care facilities and general medical practices, or deliver services such as therapeutic drug monitoring and chronic medication therapy management within community pharmacies, have the potential to ease the demands on medical practitioners. This in turn would immensely improve the access to health care.

### 4.2. Location Rules

In addition to the funding of dispensing, another CPA program relevant to community pharmacy viability is the Agreement location rules, which constrains the establishment of new pharmacies in Australia, thereby limiting competition. In effect, the rules benefit existing pharmacy proprietors above and beyond all others. In 90% of the countries of Western Europe, the locations of pharmacies are regulated using both demographic and geographical criteria [50,51]. In contrast, the absence of pharmacy location controls in the United States has led to ‘pharmacy deserts’ in lower socioeconomic communities [52,53]. If pharmacy locations are to be regulated in Australia, the process should avoid members of the public losing the opportunity to attend a pharmacy of their choice, and pharmacy proprietors should not feel immune from competition. 

The existing CPA location rules, based predominantly on the distance between pharmacies, fails to recognise specific community needs. Based on international experience, an alternate policy option to support public access to pharmacies and enhance the pharmacist services in both rural and urban ‘medically underserved areas’ could be through regulation on a health-needs basis and funding of specific pharmacist services linked to demonstrable local needs, service quality, and improvement.

### 4.3. Stakeholder Engagement

This study also challenged the inclusion of non-dispensing programs and services in the CPA. It was found to be anomalous that one pharmacy ownership organisation negotiated programs to the exclusion of those directly involved in providing the services; for example, the community service obligation, professional programs including professional services, R&D functions, and Indigenous programs. In effect, the self-interest of pharmacy owners represented by the PGA has dominated what is professed to be a public health policy that incorporates programs relevant to a diverse range of stakeholders [23]. It was suggested that the federal government engage beneficiaries and major stakeholders equitably in the development of policies.

### 4.4. International Experience

In view of the widespread concerns with the existing CPA, the examination of pharmacy policy frameworks in other countries is warranted. The United Kingdom initiated a five-year ‘contract’ between the government and approved pharmacies in 2005. An element of the initial UK contract that could be considered for funding of professional services in Australia was the classification of services into three tiers. All pharmacies were required to provide a number of base-level ‘essential’ services, while ‘advanced’ and ‘enhanced’ services were commissioned based on capability, demand, and local commissioning [54]. Adopting this structure in Australia would ensure that all pharmacies would be required to provide core professional services in order to be PBS approved, and those with the expertise and in areas of proven need could be funded to deliver an expanded range of services.

The structure and scope of subsequent UK contracts evolved to a greater extent than has occurred in the CPA [55]. In addition to incorporating new professional services such as a New Medicines Service in which pharmacies are funded for professional interventions in relation to nine clinical conditions, the UK 2019–2024 contract aligns with a national health plan, adopts a focus on medicines optimisation and safety, and incorporates a Pharmacy Quality Scheme [56]. If applied in Australia, such approaches would align with the National Health priority area; quality use of medicines and medicines safety. 

Further relevant international experience exists in the iterative evolution that occurred in three other countries during the timeframe of the CPA: Canada, NZ, and Scotland. Alberta, Canada is recognised globally as having an advanced level of pharmacists’ practices [57]. The enabling policy framework was built on a national health workforce review that supported pharmacists “*playing an increasingly important role as members of the healthcare team*” [58]., and an all-of-profession vision for pharmacists to deliver services that provide “*optimal drug therapy outcomes*” [59]. Following these national developments, the delivery of an expanded range of funded services at the provincial level, including Comprehensive Annual Care Plans, was enabled through Alberta government regulations and funding by the provincial health insurance agency, supported by the Alberta pharmacy organisations [60].

In New Zealand, a range of professional services including anticoagulant monitoring and long-term care services were proposed in a number of iterations of the profession’s aspirational National Pharmacist Services Framework [61]. The services were incorporated into the government’s Pharmacy Action Plan [62] and are now funded via an Integrated Community Pharmacy Service Agreement [63], with some being delivered on a regional needs basis. 

The Scottish government established a vision and plan for pharmacy [64], incorporating a philosophy of pharmaceutical care [65] and envisioning all pharmacists as clinical pharmacist independent prescribers. The national strategy [66] is delivered utilising an enabling structure of regional health boards [67].

Common features of these three abbreviated examples include a combination of national and local entities being engaged in the multi-party iterative development of vision statements, implementation strategies, and funding programs for clinical services, delivered in response to local health needs. This contrasts with the absence of a vision for the CPA, and the lack of consideration of the aspirational statement of professional bodies such as the PSA [68], the recommendations and obligations of state and territory governments such as the Victorian Parliamentary Inquiry into Community Pharmacy [69], and other federal departments [70], and inquiries [71].

### 4.5. International Relevance

Drawing on this research, it may be appropriate to critically appraise similar policies that underpin pharmacy practice and other aspects of health care delivery, particularly in countries with universal health insurance programs dominated by government policies. The potential for ‘regulatory capture’ by one stakeholder should be considered, particularly if the development of professional practice appears to have been impeded, as this may arise due to the vested interests of those benefiting from negotiating the current policies. 

### 4.6. Strengths and Limitations

One strength of this work was that the in-depth interviews were conducted with a broad range of very experienced individuals from all major pharmacy associations, proprietor and non-proprietor pharmacists, and significant external stakeholders. They included individuals with both federal and state responsibilities, people both supportive and critical of aspects of the profession, and consumers speaking from both personal and organisational perspectives. We have included quotations from 32 of the 38 interviewees with a total of 56 quotations, which we believe is a strength of the study. With the limited number of individuals in each category, it was not possible to analyse results by stratum. 

As an underlying aim of this research was to inform future policy development, more detailed reporting and discussions have been given to the CPA negotiation process and perceptions of the future than to historical sections. The future will be particularly relevant to young pharmacists, and a limitation of this study was the inadequate number of young interviewees to enable an analysis of that cohort’s perspectives. 

Another potential weakness is the subjective nature of qualitative research. An attempt was made to minimise bias in the reporting and analysis by the engagement of an external qualitative researcher not involved in the data collection to provide an impartial perspective on the data processing. 

## 5. Conclusions

The findings of this study support government policy enabling pharmacies to remain accessible, viable primary health facilities, whether that be via a profession-wide agreement, arbitration, pharmacy-specific contracts, or other mechanisms. Developing a future regulatory and funding framework would ideally incorporate all major stakeholders, particularly consumers, and address all aspects of the sector, recognise how pharmacists’ roles are changing, and enable greater diversity in the models of pharmacy practice. The findings also supports greater government funding for pharmacists’ expanded scopes of practice to be integrated with other government health funding programs, rather than be negotiated as part of dispensing remuneration. 

Three overarching recommendations arise from this study. The first is that consideration should be given to separating the negotiation of dispensing remuneration from the remuneration for other services. The second is the need for pharmacists to be supported by policy to work to their full scope of practice and capacity to address patient needs and enhance the quality use of medicines. The third is the need for collectively agreed, transparent, and accountable performance and quality measures to ensure the government’s expenditure on community pharmacy achieves the intended outcomes. 

Further work is necessary to iteratively develop a vision and strategy for pharmacists’ roles and to formulate more inclusive, equitable, and effective policy and funding structures to implement the strategy in Australia.

## Figures and Tables

**Table 2 pharmacy-11-00188-t002:** Socioecological strata of interviewees.

Strata	Substrata	Pharmacists	Interviewed	In Other Strata	Total in Strata
Societal	Politicians	-	2	-	
	Health bureaucrats	2	3	-	
	Pharmacy regulators	1	1	-	
	Health, economic, and political policy experts	-	3	-	
Total Societal					9
Community	Professional, industrial, and commercial pharmacy organisations	9	9	-	
	Medical, patient, and supplier organisations	1	3	-	
	Media, business, and banking commentators	-	3	-	
Total Community					15
Organisational	Pharmacy proprietors	4	4	3	
Total Organisational					7
Personal	Pharmacists working in community, hospital, academic, and consultant practice	7	7	-	
	Consumers and consumer organisations	-	3	4	
Total Personal					14
Total		24	38		

**Table 3 pharmacy-11-00188-t003:** CPA professional services listed in Agreement documents.

CPA Programs	2CPA	3CPA	4CPA	5CPA	6CPA	7CPA
Residential Medication Management Review						
Home Medication Review						
Case Conferencing						
General Medical Practitioner Facilitators						
Medication Management Review Facilitators						
Diabetes Medication Assistance Program *						
Asthma Pilot Program *						
(Patient) Medication Profiling Service *						
Emergency Hormonal Contraception *						
Diabetes Pilot Program *						
Diabetes Medication Management Service						
MedsCheck						
Clinical Interventions ^#^						
Primary Health Care ^#^						
Community Service Support ^#^						
Working with Others ^#^						
DMS						
Diabetes MedsCheck						

The shading indicates the CPA in which each program operated (excludes DAA, staged supply, continued dispensing, new start up allowance, Indigenous, QCPP, R&D, practice change, NDSS, and IT development programs). Refer to Abbreviations, for a list of abbreviations. * Components of the Better Community Health Program; ^#^ Components of the Pharmacy Practice Incentive Program.

## Data Availability

All documents used in the research are in the public domain.

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
