# Peer review of "A Qualitative Evaluation of the Australian Community Pharmacy Agreement"

_pharmacy, 2023, doi:10.3390/pharmacy11060188_

Round 1

Reviewer 1 Report

Comments and Suggestions for Authors

Summary:
The authors analyzed the experiences of different relevant stakeholders with the Community Pharmacy Agreement in Australia in view of the coming arrangements. A qualitative in depth interview method was used. The different objectives and key aspects of the agreement are presented and discussed. Suggestions are made for negotiation of future agreements.

General comments:

The topic is interesting with regard to international benchmarking of evolution of the profession but also of the governmental approach to implement health policy and vision. However, not being familiar with the Australian context I missed a more general introduction to guide the reader into the topic. I would also suggest to simplify the results and discussion by generalizing more so that the (international) reader can translate the results to other countries.

Specific comments:

Can you clarify more explicitly who is involved in the negotiation of this agreement?
When talking about the transparency, what transparency is expected, discussed: the funding, the process, the content?

Author Response

Please see the attached response

Reviewer 2 Report

Comments and Suggestions for Authors

Methods:

More explanation / text is necessary in the chapter Methods, paragraph Data Analysis; It is not clear who the second researcher is and what kind of task the second researcher has performed. Did the second researcher checked all the transcript and coding or was there random check?

Is there a reason not to include community pharmacists and patients?

Results:

Table 2 is complex. In general it is nog clear who is pharmacy (by training). And it is also unusual only mention the people who are a pharmacist. I can imagine that other professions are relevant for this research (e.g. policy advisor, economist, etc.).

·      Table 2: what is 'Other organisations' in Community? Probably an explanation is necessary.

·      Table 2: Why is 'Commentators' divided in Community? Are these people also a pharmacist by training? Probably an explanation is necessary.

In general the results of the interviewers are not clearly presented. Results are presented one after the other as a continuous story. The results are presented as an n=1 result.

The result in paragraph 3.3.1 is more clear but I miss the numbers. Is it possible to add some numbers: How many interviewers have indicated these three categories?

Conclusion:

I cannot clearly conclude the three recommendations from the results. More explanation is needed as to how the three recommendations are based.

Author Response

Please zsee the attached form

Reviewer 3 Report

Comments and Suggestions for Authors

stakeholders and not necessarily pharmacists, but adding this clarification would make the abstract more precise.

Introduction : Good. I ask me if the table 1 is very necessarily. Moreover, the number of acronyms (not defined under the table) makes it difficult to understand for an international audience.  

Known issues related by the stakeholders about the CPA are cited in a paragraph so I ask me what the study will add.

Methods : Page 4 line 122 : Table 2 (not 3)

I assume that not stakeholder from PGA were interviewed (or it may not so clear in the table 2 for a non-Australian reader like me) ? May be it would be interesting to balance the findings.

Results : I like the mix between several context sentences at the beginning of a part to introduce CPA elements, interviews findings and verbatim. This part is pleasant to read.

The table 1 reference can be found in the results section (page 11 line 418). But I will make the programs more understandable, because the acronyms (even if defined at the end of the paper) make them difficult to understand.

Table 3 is a nightmare and not understable. I suggest to see if it is very necessary or to improve understanding

Discussion

The authors spoke of “ deductive thematic analysis” : I would precise this in the method section.

The “international experience” section is interesting.

Author Response

Please see the attached form
